# Oxidative Stress Orchestrates MAPK and Nitric-Oxide Synthase Signal

**DOI:** 10.3390/ijms21228750

**Published:** 2020-11-19

**Authors:** Tsuyoshi Takata, Shoma Araki, Yukihiro Tsuchiya, Yasuo Watanabe

**Affiliations:** 1Department of Pharmacology, Showa Pharmaceutical University, Machida, Tokyo 194-8543, Japan; tsuyoshi.takata.c2@tohoku.ac.jp (T.T.); araki@ac.shoyaku.ac.jp (S.A.); yatsuchi@ac.shoyaku.ac.jp (Y.T.); 2Department of Environmental Health Sciences and Molecular Toxicology, Graduate School of Medicine, Tohoku University, Miyagi 980-8575, Japan

**Keywords:** redox regulation, nitric oxide synthase, 90-kDa ribosomal S6 kinase, Ca^2+^/calmodulin-dependent protein kinase (CaMK), phosphorylation, *S*-glutathionylation

## Abstract

Reactive oxygen species (ROS) are not only harmful to cell survival but also essential to cell signaling through cysteine-based redox switches. In fact, ROS triggers the potential activation of mitogen-activated protein kinases (MAPKs). The 90 kDa ribosomal S6 kinase 1 (RSK1), one of the downstream mediators of the MAPK pathway, is implicated in various cellular processes through phosphorylating different substrates. As such, RSK1 associates with and phosphorylates neuronal nitric oxide (NO) synthase (nNOS) at Ser847, leading to a decrease in NO generation. In addition, the RSK1 activity is sensitive to inhibition by reversible cysteine-based redox modification of its Cys223 during oxidative stress. Aside from oxidative stress, nitrosative stress also contributes to cysteine-based redox modification. Thus, the protein kinases such as Ca^2+^/calmodulin (CaM)-dependent protein kinase I (CaMKI) and II (CaMKII) that phosphorylate nNOS could be potentially regulated by cysteine-based redox modification. In this review, we focus on the role of post-translational modifications in regulating nNOS and nNOS-phosphorylating protein kinases and communication among themselves.

## 1. Introduction

The mitogen-activated protein kinases (MAPKs) are a highly conserved family of serine/threonine kinases that play a central role in the range of fundamental cellular processes like cell growth, proliferation, death, and differentiation [1]. Among them, the extracellular signal-related kinase (ERK)-1/2, c-Jun *N*-terminal kinase (JNK)-1/2/3, and p38-MAPK (α, β, δ, and γ) are most extensively studied. ERK1/2 activation controls cell proliferation. Meanwhile, the JNKs and p38 are responsive to stress stimuli and play roles in the inflammatory and the apoptotic responses, respectively [2]. The wide range of functions regulated by the MAPKs is mediated through phosphorylation of several substrates, including members of a family of protein kinases termed MAPK activated protein kinases (MAPKAPKs) [2]. In some cases, MAPK activation regulates docking interactions with MAPKAPKs, a phenomenon that was first described for the interaction between ERK1/2 and 90 kDa ribosomal S6 kinases (RSKs) family members [3]. Thus, ERK1/2 plays not only a central role in the control of cell proliferation but also an important role in numerous cellular processes in cardiomyocytes, cancer cells, and renal interstitial fibroblasts and myofibroblasts via the direct downstream effectors, RSKs [4].

RSKs contain two functional protein kinase domains, and in mammals, four expressed homologs (RSK1–4) have been identified. RSKs are activated by ERK1/2 [5] and 3-phosphoinositide-dependent protein kinase 1 (PDK1) [6] by sequential phosphorylation in the C-terminal kinase domain (CTKD) at Thr573 and the *N*-terminal kinase domain (NTKD) at Ser 221 [7], respectively (Figure 1). When activated, RSK promotes the phosphorylation of many cytosolic and nuclear targets that regulate diverse cellular processes, including cell growth, proliferation, survival, and motility [8].

Nitric oxide (NO) is a ubiquitous gaseous biologically active messenger [9]. Neuronal NO synthase (nNOS) is involved in synaptic signaling events and provides plasticity of the central nervous system (CNS) through modulating neurogenesis and CNS functions such as learning, and memory [10]. Inducible NOS is involved in cytokine-induced NO generation by various cell types including hepatocytes, macrophages, and neutrophils. It is implicated in the pathogenesis of atherosclerosis and host resistance to pathogens and contributes to the immunopathology of inflammation and septic shock [11]. Endothelial NOS (eNOS) is expressed in endothelial cells and is involved in controlling vascular functions [12], and all three NOS types have been implicated in the maintenance of redox homeostasis in the cardiovascular system [13,14].

Physiological functions of NO in the CNS include regulation of neurotransmission, neuroprotection, and neurotoxicity [15]. Initial studies indicated that NO interacts with soluble guanylyl cyclase and stimulates its activity, leading to increased cGMP. NO also regulates protein functions by redox-based posttranslational modification such as reversible *S*-nitrosylation, the addition of an NO to the thiol side chain of cysteine residues [16]. As is true for most signal transduction systems, there is cross-talk between the NO signaling and other signaling pathways. For example, targets for NO include the l-type Ca^2+^ channel and Ras, an upstream activator of MAPK [17,18,19]. We have shown earlier that NO can promote the nicotine-induced activation of MAPK using either nNOS-expressing or -non-expressing PC12 cells [20,21].

In addition to NO, reactive oxygen species (ROS) are constantly produced by several normal cellular events, with a major source being aerobic respiration. ROS are highly reactive molecular groups derived from oxygen molecules such as superoxide (O_2_^−^), hydrogen peroxide (H_2_O_2_), and hydroxyl radical (•OH), and causes oxidative stress by disturbing the redox balance in vivo [22]. In aerobic organisms, electrons leak out from the electron transport chain in the inner mitochondrial membrane at a low frequency, and oxygen directly receives them to generate superoxide [23]. Superoxide is metabolized to H_2_O_2_ by superoxide dismutase and catalase but reacts with NO produced by NOS to produce a more reactive peroxynitrite (ONOO^−^) and hydroxyl radical. Excessive ROS can induce oxidative damage in cell constituents and promote several degenerative diseases and aging [24]. Meanwhile, ROS are not only injurious to cell survival but also essential to cell signaling and regulation [25]. It has been recently noted that cells actively produce ROS by enzymes such as NADPH oxidase (NOX) and ROS can regulate the fate and function of cells [22,26]. The roles of oxidative stress in various diseases and the detection of oxidative stress biomarkers have been comprehensively reviewed in recent work [27,28]. The respiration processes in mitochondria, ROS generation, and hypoxia have also been studied using an electrochemical quartz crystal nanobalance with mitochondria immobilized on a piezoelectric quartz crystal resonator [29].

In the function of ROS as a signal molecule, the highly nucleophilic cysteine thiol plays a major role in maintaining an appropriate redox state in vivo [30]. Protein thiols form various oxidative modifications such as sulfenic acid, sulfinic acid, sulfonic acid, disulfide bonds, *S*-glutathionylation, and *S*-nitrosylation [16,30,31,32]. These thiol oxidative modifications are involved in a variety of cellular processes such as gene expression [33], energy metabolism [34,35], phosphorylation [36,37,38], and protein localization [39]. The thiol oxidized in the cell is reduced by thioredoxin, glutaredoxin (Grx), and glutathione reductase [40]. The reversibly formed protein *S*-glutathionylation is a common modification among thiol modifications. Glutathione (GSH) is abundant in cells, and sulfenic acid and *S*-nitrosylation are converted to *S*-glutathionylation, suggesting that *S*-glutathionylation is a common feature in redox signaling [41].

Several cellular stimuli that induce ROS production can activate MAPK pathways in multiple cell types [42,43]. Moreover, direct exposure of cells to exogenous H_2_O_2_ mimics oxidative stress and leads to activation of MAPK pathways [44,45]. We have shown that RSK1 was directly inhibited by *S*-glutathionylation at Cys223 [46]. These are interesting findings since ROS could activate MAPK signal and simultaneously suppress RSK1, a downstream molecule of MAPK. Furthermore, we have shown that nNOS-derived NO can promote the nicotine-induced activation of MAPK in PC12 cells [20,21]. We have also shown that epidermal growth factor (EGF)-induced RSK1 associates with and phosphorylates nNOS at Ser847, leading to inhibition of NO generation activity [47]. In other words, oxidative stress in a broad sense including NO seems to orchestrate MAPK signal and NOS signal. Aside from oxidative stress, nitrosative stress also contributes to protein *S*-glutathionylation. Thus, the protein kinases such as Ca^2+^/calmodulin (CaM)-dependent protein kinase I (CaMKI) and II (CaMKII) that phosphorylate nNOS [48,49] would be potentially regulated by *S*-glutathionylation. This article gives an overview of the role of *S*-glutathionylation in regulating NOS and NOS-phosphorylating protein kinases and in relating neuronal diseases.

## 2. The Reaction of Protein *S*-Glutathionylation

### 2.1. The Causes of Protein S-Glutathionylation

One of the key characteristics of the redox switches is the change of the redox potential. The main endogenous biomolecule defining the redox potential is the redox couple formed by GSH and its oxidized form, glutathione disulfide (GSSG), due to the high reversibility of this system and high concentration of GSH in tissues [50]. Other important molecules involved, though in much lower concentrations, are cysteine and homocysteine [51], the latter is important due to cardiotoxicity. The site of *S*-glutathionylation depends on the acid dissociation constant (pKa) of cysteine thiol. In other words, reactivity increases in a microbasic environment (low pKa) where arginine, histidine, and lysine are located nearby. Under oxidative stress, *S*-glutathionylation at these cysteine sites is caused by some reactions, such as thiol-disulfide exchange with GSSG (pathway 1), the reaction of oxidant-induced protein sulfenylation (P-SOH) with GSH (pathway 2), the reaction of NO-induced *S*-nitrosylation protein (P-SNO) with GSH (pathway 3), and reaction with *S*-nitrosoglutathione (pathway 4) (Figure 2). The cell-permeable diamide, a thiol group-selective oxidant, facilitate *S*-glutathionylation formation under mimicking oxidative (pathway 2) or nitrosative (pathway 3) stress [52]. That is, the post-translational *S*-glutathionylation is an established modification as a response against ROS/ reactive nitrogen species (RNS) and *S*-nitrosylation and *S*-glutathionylation may occur at the same site. In fact, we have found that CaMK I activity is inhibited by either *S*-nitrosylation or *S*-glutathionylation at Cys179 [53,54]. Meanwhile, CaMKII is sensitive to inhibition by *S*-nitrosylation of Cys6 residues and is also sensitive to inhibition by *S*-glutathionylation of the residues besides Cys6 [54].

### 2.2. The Biological Meaning of Protein S-Glutathionylation

*S*-glutathionylation plays an important role in protecting cysteine residues from irreversible oxidation during oxidative stress. Meanwhile, it can change the structure, function, and activities of modified proteins which affects signal transduction. Both the forward- (*S*-glutathionylation) and reverse- (deglutathionylation) reactions are involved in the pivotal redox-dependent cell signaling cascades. Grx is an essential thioltransferase whose primary role is to reduces *S*-glutathionylation. Grx 1 insufficiency results in increased levels of *S*-glutathionylation, implicated in many pathological events, including cardiovascular [56] and Parkinson’s [57] diseases. Since Grx catalyzes GSH-dependent reduction of *S*-glutathionylation, lower GSH:GSSG ratio might attenuate Grx activity, mimicking Grx deficiency. In contrast, Grx 1 plays a primary proinflammatory role in microglia [57] or an anti-angiogenic role in the vasculature [58]. More than one hundred *S*-glutathionylation proteins have been experimentally verified. Thus, *S*-glutathionylation is thought to participate in the protection and progression of pathological consequences, requiring examination of cell-type specific manipulation of Grx1 content or activity.

### 2.3. Identification of Modified Molecules

The direct and indirect methods are employed to identify the modified molecules. As a direct method, technically advanced proteomics have facilitated high-level enhanced sensitivity of *S*-glutathionylation detection [59]. There are other direct methods based on the use of anti-GSH antibody and biotinylated GSH or membrane-permeable biotinylated reduced GSH ethyl esters. The former is convenient because it is one of the few antibodies capable of detecting protein oxidative modification. The latter can concentrate modified proteins by avidin-biotin binding. However, the effect of exogenous biotinylated GSH itself on the endogenous redox state must be considered. In the indirect method, the free cysteine thiol on the *S*-glutathionylated protein is blocked with an alkylating agent and then recovered with a resin having an affinity for the thiol obtained by reducing *S*-glutathionylation. However, it is important to ensure the selectivity of de-*S*-glutathionylation by reductive treatment. Bioinformatics has also predicted the site of protein cysteine residues that are *S*-glutathionylated [60].

## 3. Function of *S*-Glutathionylation

Oxidant-induced protein sulfenylation (PS-OH) is unstable, and excessive oxidation results in irreversible modification of sulfinylation to sulfonylation. Sulfenic acid can be also further oxidized to form disulfide bonds, *S*-glutathionylation (Figure 2). Therefore, the protection of cysteine thiol from irreversible oxidative modification is affected by the spatiotemporal amount of intracellular GSH. *S*-glutathionylation responds to oxidative and nitrogen stresses, and there is coordination among *S*-nitrosylation, *S*-glutathionylation, and phosphorylation in the regulation of the NO signaling system.

### 3.1. Endothelial and Neuronal NOSs

The *S*-glutathionylation of eNOS was reported in the vascular wall of hypertensive rats [61] and endothelial cells exposed to either hypoxia/reoxygenation [62] or decreasing GSH biosynthesis condition [63]. The *S*-glutathionylation of eNOS occurs on Cys689 and Cys908, which are critical to maintaining the normal function of the enzyme [61]. Consequently, *S*-glutathionylation triggers eNOS uncoupling, switching the protein activity from an NO producing enzyme towards a NOX activity producing O_2_^−^, causing the pathophysiology of cardiovascular and renal diseases [61,64,65,66,67]. Another major consequence is that the decreased NO production by *S*-glutathionylation of eNOS is implicated in the pathogenesis of necrotizing enterocolitis (NEC). The eNOS-derived NO is inhibited by *S*-glutathionylation of eNOS, which can be reversed by Grx1. Grx1 deficient mice increase the severity of experimental NEC in association with *S*-glutathionylation of eNOS [68]. Thus, the uncoupling reaction of eNOS due to site-specific *S*-glutathionylation may contribute to endothelial dysfunction in various pathological conditions.

The zinc-tetrathionate motif present in either nNOS or eNOS is related to enzyme activity. The Cys96 and Cys101, which comprise the eNOS zinc-tetrathiolate complex, have been identified as the key sites of eNOS *S*-nitrosylation [69,70]. Cys331, one of the zinc-tetrathiolate cysteines, was identified as the key site of nNOS *S*-nitrosylation [71,72]. In addition, it was reported that nNOS is highly *S*-nitrosylated in resting rat hippocampal neurons and the enzyme undergoes de-nitrosylation during the process of rat brain ischemia/reperfusion [73]. Interestingly, the process of nNOS de-nitrosylation is coupling with the decrease of nNOS phosphorylation at Ser847, a site associated with nNOS activation [74].

### 3.2. RSK1

Activation of RSK1 is regulated by its upstream kinases, ERK1/2 and PDK1, by sequential phosphorylations. Sequential phosphorylations of RSK1 are initiated by ERK1/2 at Thr573 of the CTKD leading to the autophosphorylation of RSK1 at Ser380. This phosphorylation allows the dockage of PDK1 and enables PDK1-dependent phosphorylation in the NTKD of RSK1 at Ser221, resulting in its maximal activation [5,7] (Figure 1). The Cys in the subdomain VIII, often mentioned as a key protein kinase catalytic domain indicator [75], is highly conserved among several kinases including RSK1 (Cys223 in the NTKD, Cys575 in the CTKD: based on the mouse RSK1 sequence). We previously demonstrated that Cys223 in the NTKD, but not Cys575 in the NTKD are the *S*-glutathionylation-sensitive site to regulate the activity of RSK1 [46]. The EGF-induced activation of RSK1 activity by phosphorylation at Ser221 was inhibited by post-treatment with oxidative stress (diamide/GSH) in vitro. Thus, inhibition of RSK1 by its *S*-glutathionylation at Cys223 appears to be dominant over activation of the kinase by its phosphorylation at Ser221 (Figure 3).

We reported that nNOS is phosphorylated by EGF-induced activated RSK1, leading to a reduction of NO producing activity through the phosphorylation at Ser847 of the enzyme in cells [47] (Figure 3). The phosphorylation of ERK1/2 was increased similarly following treatment with either EGF or diamide in cells [46]. That consistent with reports showing that oxidative stimulation increases phosphorylation of ERK1/2 through oxidative modifications of MAPK signaling via inactivation of MAPK phosphatases by oxidation [77]. Treatment of the cells expressing nNOS with EGF but not diamide resulted in an increase in the phosphorylation of nNOS at Ser847. RSK1-induced nNOS phosphorylation at Ser847 was inhibited by the post-treatment of diamide in situ, indicating that oxidative stress might up-regulate the NO production via RSK1 inactivation. Thus, a different mechanism of ERK activation by either EGF or oxidative stress was regulated via RSK1 activation/inhibition in cells (Figure 3). RSK1 is a central mediator of ERK during cell survival [78]. EGF could simply activate the ERK-RSK1 signal but oxidative stress might activate ERK and inactivate RSK1 simultaneously. On the other hand, it has also been reported that the binding of RSK to nNOS occurs in the solitary tract nucleus, which induces activation of nNOS through the phosphorylation at Ser1416 of the enzyme, and the ERK1/2-RSK1-nNOS signaling pathway is involved in the brainstem control of blood pressure in hypertensive rats by angiotensin II [79].

We further examined another Cys223-sensitive redox signaling aside from *S*-glutathionylation. Although RSK1 activity was insensitive against several NO donors, it was inhibited by H_2_O_2_ (our unpublished observation). It was recently demonstrated that a large portion of cysteine thiols are oxidized to persulfidated thiols such as cysteine perthiosulfenic acid (CysSSOH) and perthio-glutathion (CysSSSG) [80]. Quantitative data indicated that hydropersulfides and inorganic polysulfides are widespread in cells and tissues and occur at much higher physiological concentrations than ROS or RNS [81,82]. In recent years, it was revealed that extensive and prevalent cysteine polysulfidation was introduced co-translationally, sustained in the mature protein, and physiologically present even in the post-translational processes of the cells [83]. Thus, polysulfidation may be a more common regulatory mechanism than *S*-glutathionylation and *S*-nitrosylation. In fact, RSK1was inactivated by its polysulfidation at Cys223 residue (our unpublished observation). The RSK1 inactivation by *S*-glutathionylation, *S*-oxidation, and polysulfidation at Cys223 might play important roles in regulating NOS signaling (Table 1).

### 3.3. CaMKI

Multifunctional CaMKs are activated by the phosphorylation of a crucial threonine residue either by itself (CaMKII) or by upstream kinases, CaMK kinases (CaMKKs) (CaMKI and CaMKIV). Multifunctional CaMKs, present in most mammalian tissues, can phosphorylate many downstream targets, thereby regulating numerous cellular functions. Aside from canonical post-translational modifications, cysteine-based redox switches in CaMKs affect their enzyme activities [84]. It was shown that multifunctional CaMKs activities were directly regulated by cysteine-based redox switches [54].

We reported previously CaMKI is inhibited by *S*-glutathionylation, *S*-nitrosylation, or *S*-oxidation of Cys179 in the activation loop [53,54] (Figure 4). This Cys179 is an equivalent activation loop Cys residue in the subdomain VIII [75]. We recently reported that CaMKI was also inactivated by its polysulfidation at Cys179 residue, leading to being protected from irreversible oxidative modifications [85]. Inhibition of CaMKI by oxidation appears to be dominant over activation of the kinase by phosphorylation at Thr177 [53,85]. We reported previously that CaMKI phosphorylates at Ser741, leading to a reduction of nNOS activity by blocking the binding of Ca^2+^/CaM to nNOS [48] (Figure 4). The CaMKI inactivation by *S*-glutathionylation, *S*-nitrosylation, *S*-oxidation, or polysulfidation at Cys179 might play important roles in regulating NOS signaling (Table 1).

### 3.4. CaMKII

It was reported that oxidized CaMKIIδ at Met281/282 in cardiomyocytes promotes mortality after myocardial infarction [86,87] and contributes to ischemia/reperfusion injury in the heart [88]. *S*-nitrosylation of Cys280 and Cys289 (α), Cys290 (β, γ, and δ) also induce autonomous activity [89,90]. Prior *S*-nitrosylation of CaMKIIδ at Cys273, which is not conserved in CaMKIIα, to Ca^2+^/CaM exposure strongly suppresses kinase activation [90]. In addition to these cysteine-based redox switches, we reported that glutamate induces Cys6 (and/or) Cys30 dependent *S*-nitrosylation of CaMKIIα in the hippocampal slices [91]. *S*-nitrosylation of Cys6 inhibits CaMKIIα activity in ATP competitive fashion. Furthermore, we showed that *S*-glutathionylation induced by diamide/GSH also inhibits CaMKIIα activity, but it is not via the site-specific modification at Cys6 [54]. We also recently reported that CaMKII was also inactivated by its polysulfidation at Cys6 residue [92]. Phosphorylation of nNOS at Ser847 by CaMKII decreases NO generation and increases superoxide generation [76]. Thus, CaMKII activity is regulated by multiple sites of oxidation excluding *S*-glutathionylation (Table 1).

### 3.5. CaMKIV

It has been reported that CaMKIV activity is increased through the inactivation of protein phosphatases in oxidative cellular environments [38]; however, the direct mechanism involving CaMKIV remained unknown. We recently reported that CaMKIV is inactivated by polysulfidation of the active-site Cys198 residue [93]. Our data indicate that Cys198 in CaMKIV represents a major target of endoplasmic reticulum (ER) stress signaling and exogenously applied inorganic polysulfides and that polysulfidation at this site inhibits the enzyme activity. We also recently observed that the inhibition of CaMKIV by *S*-oxidation at Cys198 was observed following extracellular treatment with H_2_O_2_ [94]. Treatment of recombinant CaMKIV with diamide, a thiol oxidizing agent, either in the presence or absence of GSH results in inactivation of the enzyme, and mutated CaMKIV (C198V) was refractory to the inhibition. In addition, CaMKIV is not inhibited by NO unlike CaMKI and CaMKII [54]. CaMKIV is inhibited via its polysulfidation and *S*-oxidation at Cys198 but neither its *S*-glutathionylation nor its *S*-nitrosylation. Thus, polysulfidation and *S*-oxidation but not *S*-glutathionylation may serve to mediate CaMKIV inactivation as a response to ER stress (Table 1).

## 4. Roles of Protein *S*-Glutathionylation in Diseases

The brain is highly susceptible to oxidative stress due to its requirement for 20% of the total basal oxygen budget to support ATP intensive neuronal activity [95], as well as its inability to undergo cellular regeneration. This is largely related to the pathogenesis of oxidative stress with aging as a risk factor for neurodegenerative diseases. Indeed, various indices of ROS damage have been reported within the specific brain region that undergoes selective neurodegeneration [96]. Roles of RSKs in kidney disease, cardiovascular disorders, and cancers and protein glutathionylation in neurodegenerative diseases have been extensively reviewed [4,97].

### 4.1. Neurotoxicity with Increased NO through Redox Switch of RSK1/CaMKI/CaMKII

Inhibition of NOS is neuroprotective, implicating NOS in the production of radicals which can produce peroxynitrite and H_2_O_2_ [98]. Following EGF treatment, nNOS was phosphorylated by RSK1 at Ser847 in rat hippocampal neurons and cerebellar granule cells, which suggests a novel role for RSK1 in the regulation of NO function in the brain [47]. It was shown that nNOS is phosphorylated at Ser847 by CaMKII, leading to neuroprotective effects against cerebral ischemia/reperfusion injury [74]. RSK1 is present in the brain and could be responsible for phosphorylation of nNOS at Ser847, other than CaMKII.

The CaMKK-CaMKI cascade is a significant player in neuronal development including neurite outgrowth [99]. CaMKI is involved in neuronal development in NG108 cells [100] and cultured hippocampal neurons [101] expressing nNOS. A constitutively active form of CaMKI causes abnormal morphology when transfected into Hela cells [102] not expressing nNOS. Thus, redox-sensitive inhibition of CaMKI through the modification of Cys179 residue may be required for normal neuronal development. Conversely, it was shown that nNOS is phosphorylated by CaMKI, leading to a reduction of nNOS activity through the phosphorylation at Ser741 of the enzyme [48]. Since CaMKI-induced nNOS phosphorylation at Ser741 was inhibited by the treatment of diamide in situ [53], oxidant stress might up-regulate the production of NO via CaMKI inactivation. CaMKI promoted mitochondrial fission via phosphorylation of dynamin-related protein 1 (Drp1), a mediator of mitochondrial fission [103]. *S*-nitrosylation of Drp1 is increased in the brains of human Alzheimer’s disease patients and mediates beta-amyloid-related mitochondrial fission and neuronal injury [104]. Mitochondria undergo rapid fragmentation with a concomitant increase in ROS formation after exposure to high glucose concentrations [105]. Oxidative stress at least in part might inhibit mitochondrial fission via CaMKI inactivation.

### 4.2. Huntington’s Disease through Redox Switch of RSK

Huntington’s disease (HD) is an autosomal dominant disease that causes degeneration of medium spiny γ-aminobutyric acid neurons, main symptoms are involuntary movements and psychiatric symptoms/dementia caused by the addition of polyglutamine to the *N*-terminal of a gene product called Huntingtin. HD aggregated the Huntington protein with expanded polyglutamine repeats forms inclusions in the nucleus [106]. Furthermore, increases in GSH, Grx, and GSH reductase were detected in the brains of transgenic HD mice models [107,108]. *S*-glutathionylation of transient receptor protein C5, which is one of the calcium channels, was found in the postmortem striatum, suggesting cell death by the Ca^2+^/CaM-dependent signaling system due to intracellular Ca^2+^ influx [109]. The expression levels of and the activities of RSK1/RSK2 are increased in the striatum of HD animal models [110]. Thus, the inhibition of RSK1 activity by *S*-glutathionylation may exacerbate the pathology.

## 5. Conclusions

We propose here the phosphorylation- and NOS-signaling are mutually regulated via phospho-nNOS at Ser741/*S*-glutathionylated CaMKI at Cys179 and phospho-nNOS at Ser847/*S*-glutathionylated RSK1 at Cys223 and *S*-nitrosylated CaMKII at Cys6/30. In more detail, the CaMKI activity is inhibited via its *S*-glutathionylation, *S*-nitrosylation, *S*-oxidation, or polysulfidation of Cys179. The CaMKII activity is inhibited via its polysulfidation and *S*-nitrosylation at Cys6 but neither its *S*-glutathionylation nor *S*-oxidation at Cys6. The RSK1 activity is inhibited via its *S*-glutathionylation, *S*-oxidation, and polysulfidation at Cys223 but not by its *S*-nitrosylation (Table 1). Thus, CaMKI is most susceptible to cysteine-based redox switches. The modification of Cys residue in CaMKI (Cys179), RSK1 (Cys223), or CaMKIV (Cys198) is located in the subdomain VIII [75] and might be regulated by a similar type of modification as *S*-glutathionylation or *S*-oxidation. The CaMKIV activity is inhibited via its polysulfidation and *S*-oxidation at Cys198 but neither its *S*-glutathionylation nor its *S*-nitrosylation. CaMKIV is localized predominantly in the nucleus and cannot phosphorylate nNOS which is not present in the nucleus and might not be sensitive to nitrosative stress. However, it is not clear how nNOS-phosphorylating protein kinases as CaMKI, CaMKII, and RSK1 are regulated by the specific type of redox-sensitive Cys modification in pathophysiological response in cells. We have shown that NO can promote the nicotine-induced activation of MAPK in PC12 cells [20,21] (Figure 3). The enhancement of nicotine-induced MAPK activation by NO was induced by GSH depletion and was inhibited upon GSH up-regulation in cells [20]. In the condition of GSH up-regulation, *S*-glutathionylated proteins were increased in cell lysates, suggesting *S*-glutathionylation might not be involved in NO-induced promotion of MAPK activation by nicotine [20]. The target molecule and modification type of NO need to be elucidated. It was suggested that EGF induced NOX-mediated H_2_O_2_ production which induced oxidation of and inhibition of protein Tyr phosphatases, leading to enhanced EGF receptor activation [111]. The calcium-sensitive NOX, dual oxidase 1 was identified as the enzymatic source of EGF-stimulated H_2_O_2_ production in A431 and HaCaT cells [112]. It appears that NOX1, NOX2, and NOX4 are present in neurons, astrocytes, and microglia, and at least under some circumstances, NOX expression in certain CNS regions might be inducible, rather than constitutive [113]. Thus, oxidative stress could orchestrate MAPK and Ca^2+^-signals including CaMKs and NOS activations to modulate diverse neuronal processes.

## Figures and Tables

**Figure 1 ijms-21-08750-f001:**
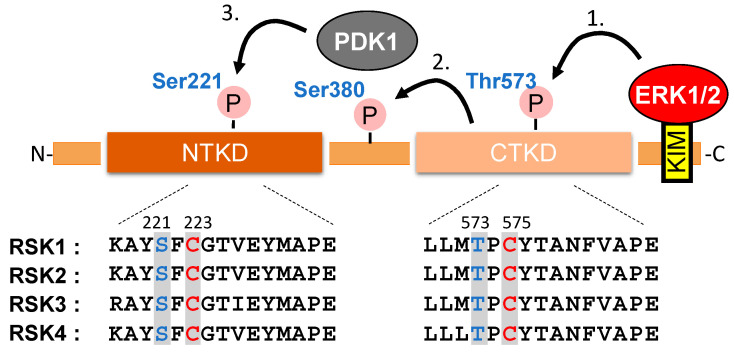
Schematic representation of the major functional domains and activation mechanism of RSK1. The *C*-terminal end of RSK1-4 contains an ERK1/2-docking domain resembling a KIM (kinase interaction motif), and the extreme *C*-terminus comprises a PDZ-binding motif. PDZ is an initialism combining the first letters of the first three proteins discovered to share the domain—post synaptic density protein (PSD95), Drosophila disc large tumor suppressor (Dlg1), and zonula occludens-1 protein (zo-1). (1) Activation of RSK is initiated by ERK docking, which is followed by the phosphorylation at Thr573 of the *C*-terminal kinase domain (CTKD) [5]. (2) The active CTKD phosphorylates Ser380 on a linker site between the kinase domains that creates a docking site for PDK1 [6]. (3) In the end, PDK1 activates the *N*-terminal kinase domain (NTKD) by phosphorylation at Ser221 [7]. The positions corresponding to functionally critical phosphorylation sites are highlighted in blue and reactive cysteine residue in the equivalent catalytic subdomain of several protein kinases are highlighted in red.

**Figure 2 ijms-21-08750-f002:**
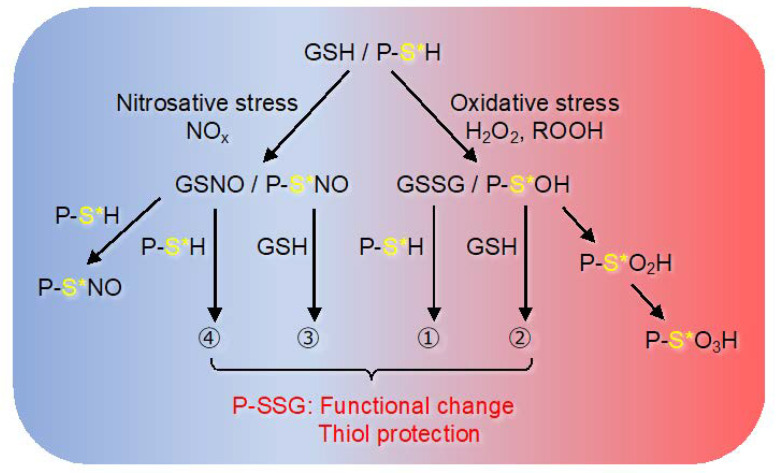
Oxidative/nitrosative stress-induced protein *S*-glutathionylation. This figure depicts oxidative/nitrosative stress-induced various mechanisms by which protein thiol moieties could be converted to *S*-glutathionylated protein (PSSG). (1) via thiol-disulfide exchange with GSSG; (2) via protein sulfenylation (P-SOH); (3) via protein *S*-nitrosylation (P-SNO); (4) via reaction with *S*-nitrosoglutathione (GSNO). (See the Section 2.1 in text for further explanation). Protein sulfenylation (P-SOH) is further oxidized to generate protein sulfinylation (P-SO_2_H) and protein sulfonylation (P-SO_3_H). The former is reversible and the latter is irreversible terminal modifications. Some proteins are both *S*-glutathionylated and *S*-nitrosylated by reaction with GSNO [55]. Asterisks indicate reactive cysteine thiols in a microbasic environment (low pKa) near arginine, histidine, and lysine. The figure is an image of the nitrosative (left part, shown by the blue background) and the oxidative (right part, shown by the red background) conditions.

**Figure 3 ijms-21-08750-f003:**
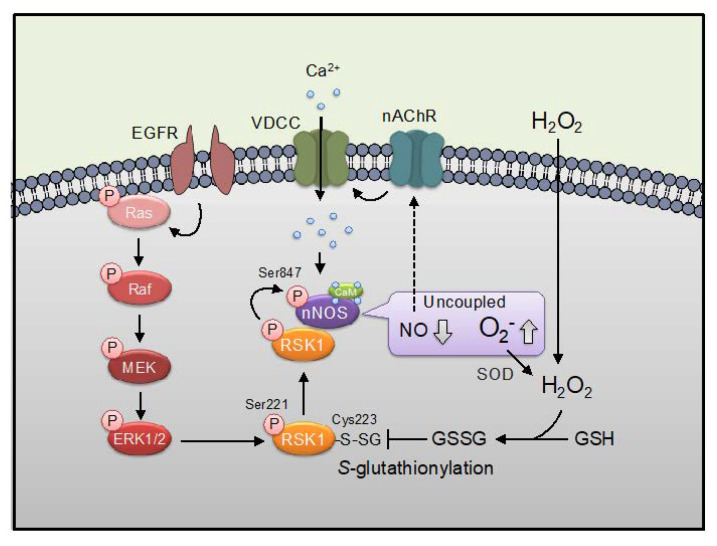
Harmonization of Ca^2+^, EGF, and oxidative signals. Ca^2+^ influx induced by voltage-dependent calcium channel (VDCC) via nicotinic acetylcholine receptor (nAChR) activates nNOS, leading NO production. MAPK signaling (Ras/Raf/MEK/ERK) induced by EGF stimuli evokes activation of RSK1 signaling which induces phosphorylation at Ser847 of nNOS. Phosphorylation at Ser847 switches nNOS reaction from NO synthesis to superoxide (O_2_^−^) synthesis via induction of uncoupling of nNOS [76] Superoxide is converted to H_2_O_2_ by superoxide dismutase (SOD). Oxidative stimulation such as H_2_O_2_ and diamide inhibit RSK1 activity via *S*-glutathionylation at Cys223 of RSK1. nNOS-derived NO can promote the nicotine-induced activation of MAPK (shown by a dashed arrow) [20,21].

**Figure 4 ijms-21-08750-f004:**
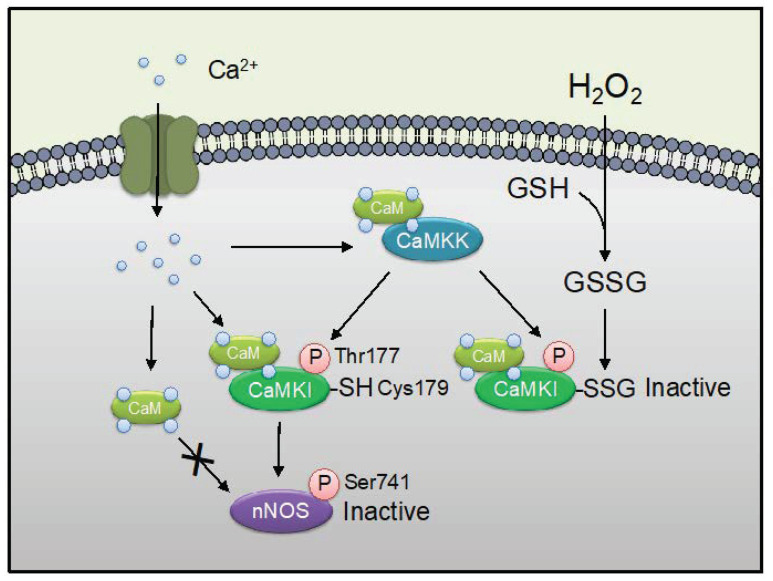
The *S*-glutathionylation of CaMKI modulates nNOS enzyme activity. Ca^2+^ influx activates CaMKK, CaMKI, and nNOS. CaMKI is activated by Ca^2+^/CaM binding and phosphorylation at Thr177 by upstream CaMKK. Activated CaMKI phosphorylates at Ser741 of nNOS in the CaM binding region, as a result, CaM is not to be able to bind to nNOS. H_2_O_2_ stimuli indices *S*-glutathionylation at Cys179 nearby Thr177 phosphorylation site of CaMKI. Note that *S*-glutathionylation of CaMKI at Cys179 appeared to be dominant over activation of the kinase by phosphorylation at Thr177 [53,83]. Thus, the illustration means ROS signaling mediates NO signaling via *S*-glutathionylation of CaMKI.

**Table 1 ijms-21-08750-t001:** The types of redox-sensitive Cys modification in protein kinases.

Type	Site	S-NO	S-SG	S-Ox	S-SH	Mechanisms
CaMKI	Cys179	+	+	+	+	Inhibition with decline of Vmax
CaMKII	Cys6	+	−	−	+	Inhibition with ATP competitive fashion
CaMKIV	Cys198	−	−	+	+	Inhibition of phosphorylation at Thr196
RSK1	Cys223	−	+	+	+	Inhibition with decline of Vmax

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
