# Peer review of "Oxidative Stress Orchestrates MAPK and Nitric-Oxide Synthase Signal"

_ijms, 2020, doi:10.3390/ijms21228750_

Round 1

Reviewer 1 Report

The manuscipt by Takata et al.:" Oxidative stress orchestrates MAPK and nitric-oxide 3 synthase signal" descbribe the influence of the oxidative stress on the post translational modifications and functions of by the members of the MAPK kinases.

The strong points of the review are:

-well described and less known modifications that modulate MAPK kinase activity

-nice illustrations depicting mode of action and the crossing of the pathways

Some minor points:

  1. Why the Authors decided to put the Figures away from the text and as a part of Introduction, when they are being reffered to later in the text? This is not intuitive for the Reader.
  2. Could the Authors perhaps could also illustrate proposed mechanisms in the mentioned diseases?? It would make the take home message much easier.

Author Response

Some minor points:

Point 1: Why the Authors decided to put the Figures away from the text and as a part of Introduction, when they are being reffered to later in the text? This is not intuitive for the Reader.

Response 1: As reviewer’s comment, we have cited the Figures and Tables in correct position.

Point 2: Could the Authors perhaps could also illustrate proposed mechanisms in the mentioned diseases?? It would make the take home message much easier.

Response 2: As reviewer’s comment, we have illustrated proposed mechanisms in graphic abstract.

Reviewer 2 Report

This interesting review focuses on various aspects of cell signaling, induced by the reactive oxygen species (ROS) and reactive nitrogen species (RNS), which is essential to trigger the activation of mitogen-activated protein kinases (MAPKs). These kinases control the phosphorylation of neuronal nitric oxide (NO) synthase (nNOS), S-glutathionylation of ribosomal S6 kinase 1 (RSK1), and Ca2+/calmodulin (CaM)-dependent protein kinases I and II (CaMKI, CaMKII), that phosphorylate nNOS. Since MAPK kinases play the key role in fundamental cellular processes, including cell growth, proliferation, apoptosis, and differentiation, the consequences of the oxidative stress imposed by ROS and/or RNS on cells are far reaching and may results in serious diseases of all organs in the body. In this review, particular attention has been paid to the role of S-glutathionylation in regulating NOS and NOS-phosphorylating protein kinases, involved in neuronal diseases.

I recommend the paper for publication after minor revision addressing the issues listed below.

  1. The Subsection numbers “3.3. .” and following - delete the last dot. The titles of Subsections 4.1., 4.2. – should not end with a dot.
  2. The roles of the oxidative stress in various diseases and the detection of oxidative stress biomarkers have been comprehensively reviewed in recent work (2012, “Detection of Oxidative Stress Biomarkers Using Functional Gold Nanoparticles”, in E. Matijevic (ed), Fine Particles in Medicine and Pharmacy, Springer Sci Publ., New York pp 241-281; ISBN: 978-1-4614-0378-4) and also in book devoted to oxidative stress (2011. “DNA-protective mechanisms of antioxidant glutathione intervention in catechol-mediated oxidative DNA damage in the presence of copper(II) ions” in S. Andreescu and M. Hepel (eds) Oxidative Stress: Diagnostics, Prevention and Therapy, Oxford University Press, Washington pp 177-209). The respiration processes in mitochondria, ROS generation, and hypoxia have also been studied using an electrochemical quartz crystal nanobalance with mitochondria immobilized on a piezoelectric quartz crystal resonator (Biosens Bioelectron. 2017;88:114-121; doi: 10.1016/j.bios.2016.07.110). These relevant studies should be cited.
  3. One of the key characteristics of the redox switches is the change of the redox potential. The main endogenous biomolecule defining the redox potential is the redox couple formed by glutathione (GSH) and its oxidized form, glutathione disulfide (GSSG), due to the high reversibility of this system and high concentration of GSH in tissues (J Photochem Photobiol A: Chemistry. 2011;225:72–80; doi: 10.1016/j.jphotochem.2011.09.028). Other important molecules involved, though in much lower concentrations, are cysteine and homocysteine (Biophys Chem. 2010;146:98-107), the later is important due to cardiotoxicity.
  4. There are some typographical and English errors which should be corrected:

Line 13: “are” – should be “is”;

Line 15: “Meanwhile, RSK1 is …” – change to: “In addition, the RSK1 activity is …”;

Line 19: “those … would be” – change to: “that … could be”;

Line 68: “causes oxidative stress via disturb the balance of redox” – should be “cause the oxidative stress by disturbing the redox balance”;

Line 76: “and regulates” – change to: “and ROS can regulate”;

Line 98: “those” – should be “that” (meaning: “which”, no plural);

Line 344: “modification Cys” – should be “modification of Cys”;

Line 353: “activated in the condition of” – change to “induced by”;

Line 354: “was inhibited in the condition of GSH up-regulation” – change to “was inhibited upon GSH up-regulation”.

Author Response

Point 1: The Subsection numbers “3.3. .” and following - delete the last dot. The titles of Subsections 4.1., 4.2. – should not end with a dot.

Response 1: As reviewer’s comment, we have amended them (see lanes 262, 325, and 348, highlighted in red).

Point 2: The roles of the oxidative stress in various diseases and the detection of oxidative stress biomarkers have been comprehensively reviewed in recent work (2012, “Detection of Oxidative Stress Biomarkers Using Functional Gold Nanoparticles”, in E. Matijevic (ed), Fine Particles in Medicine and Pharmacy, Springer Sci Publ., New York pp 241-281; ISBN: 978-1-4614-0378-4) and also in book devoted to oxidative stress (2011. “DNA-protective mechanisms of antioxidant glutathione intervention in catechol-mediated oxidative DNA damage in the presence of copper(II) ions” in S. Andreescu and M. Hepel (eds) Oxidative Stress: Diagnostics, Prevention and Therapy, Oxford University Press, Washington pp 177-209). The respiration processes in mitochondria, ROS generation, and hypoxia have also been studied using an electrochemical quartz crystal nanobalance with mitochondria immobilized on a piezoelectric quartz crystal resonator (Biosens Bioelectron. 2017;88:114-121; doi: 10.1016/j.bios.2016.07.110). These relevant studies should be cited.

Response 2: As reviewer’s comment, we have cited the papers as below (see lanes 78-82, highlighted in red).

The roles of the oxidative stress in various diseases and the detection of oxidative stress biomarkers have been comprehensively reviewed in recent work [27, 28]. The respiration processes in mitochondria, ROS generation, and hypoxia have also been studied using an electrochemical quartz crystal nanobalance with mitochondria immobilized on a piezoelectric quartz crystal resonator [29].

Point 3: One of the key characteristics of the redox switches is the change of the redox potential. The main endogenous biomolecule defining the redox potential is the redox couple formed by glutathione (GSH) and its oxidized form, glutathione disulfide (GSSG), due to the high reversibility of this system and high concentration of GSH in tissues (J Photochem Photobiol A: Chemistry. 2011;225:72–80; doi: 10.1016/j.jphotochem.2011.09.028). Other important molecules involved, though in much lower concentrations, are cysteine and homocysteine (Biophys Chem. 2010;146:98-107), the latter is important due to cardiotoxicity.

Response 3: As reviewer’s comment, we have added the sentences as below (see lanes 119-123, highlighted in red).

One of the key characteristics of the redox switches is the change of the redox potential. The main endogenous biomolecule defining the redox potential is the redox couple formed by glutathione (GSH) and its oxidized form, glutathione disulfide (GSSG), due to the high reversibility of this system and high concentration of GSH in tissues [50]. Other important molecules involved, though in much lower concentrations, are cysteine and homocysteine [51], the latter is important due to cardiotoxicity.

There are some typographical and English errors which should be corrected:

Line 13: “are” – should be “is”;

Response: As reviewer’s comment, we have changed to “is” (see lane 13, highlighted in red).

Line 15: “Meanwhile, RSK1 is …” – change to: “In addition, the RSK1 activity is …”;

Response: As reviewer’s comment, we have changed to “In addition, the RSK1 activity is…” (see lanes 15-16 , highlighted in red).

Line 19: “those … would be” – change to: “that … could be”;

Response: As reviewer’s comment, we have changed to “that … could be” (see lane 19, highlighted in red).

Line 68: “causes oxidative stress via disturb the balance of redox” – should be “cause the oxidative stress by disturbing the redox balance”;

Response: As reviewer’s comment, we have changed to “cause the oxidative stress by disturbing the redox balance” (see lane 69, highlighted in red).

Line 76: “and regulates” – change to: “and ROS can regulate”;

Response: As reviewer’s comment, we have changed to “and ROS can regulate” (see lane 77, highlighted in red).

Line 98: “those” – should be “that” (meaning: “which”, no plural);

Response: As reviewer’s comment, we have changed to “that” (see lane 104, highlighted in red).

Line 344: “modification Cys” – should be “modification of Cys”;

Response: As reviewer’s comment, we have changed to “modification of Cys” (see lane 368, highlighted in red).

Line 353: “activated in the condition of” – change to “induced by”;

Response: As reviewer’s comment, we have changed to “induced by” (see lane 378, highlighted in red).

Line 354: “was inhibited in the condition of GSH up-regulation” – change to “was inhibited upon GSH up-regulation”.

Response: As reviewer’s comment, we have changed to “was inhibited upon GSH up-regulation” (see lane 378, highlighted in red).

Reviewer 3 Report

The manuscript ‘Oxidative stress orchestrates MAPK and nitric-oxide synthase signal’ by Tsuyoshi Takata, Shoma Araki, Yukihiro Tsuchiya and Yasuo Watanabe fully describes the role of post-translational modifications in regulating nNOS. The review is complete, well written, and well organized. However, I feel that some improvements can be made in order to make the text easier for the reader to appreciate.

Major remarks

- The abstract is not straightforward on the focus of the review. The main topic of the review seems to be the relevance of redox-sensitive modifications of proteins involved in MAPK and NOS signaling, but this is not evident in the abstract, since the leading sentences introduce ROS , then MAPKs and RSK1 regulation of NOS, and only then (line 16) S-glutathionylation is finally mentioned. The focus of the review would be much clearer if the introductionary sentences would describe the cysteine-based redox switches in primis, and then their role in regulation of MAPKs and NOS signaling.

- Chapter 2 would be more useful if it could explain the biological meaning of protein S-glutathionylation, its impact in biology (how many proteins have been found to undergo this kind of modification?) and then the details of how it happens and how it is investigated. The focus of the review is the modification of NOS and CAMKs, but a general introduction to this would help contextualize the topic in the background knowledge on this kind of modification.

- Figure 2 should indicate the difference among sulfenylation, sulfinylation and sulfonylation (label the modifications).

- Figures should be cited in the text but not in the subtitles. Why are the Figures and Tables in a separate section?

Minor revisions

Line 19. What ‘those’ stands for?

Lines 32-33. This affirmation is quite general and sincerely redundant. Either the reader do not know MAK cascades, and the authors should explain them far better, or the authors assume the reader already know what is the matter, and this affirmation is not necessary at all.

Line 45. A space needs to be removed

Line 55. Why ‘despite’?

Line 56. A new paragraph should be initiated

Line 68. Grammar needs adjustment

Line 98. That not those

Line 170. Shouldn’t Table I be cited here?

Line 188. O2- is mistyped

Line 227. ‘RSKs is’ should be either ‘RSK1 is’ or ‘RSKs are’.

Line 234, it ‘was inhibited’.

Line 246. Title needs fixing

Line 273. Cys6 is mistyped.

Line 292. The title is not explicative enough of the focus of this paragraph.

Line 293. Who defines when glucose usage is ‘excessive’?

Line 343. ‘polysulfidation’ is mistyped

Author Response

Major remarks

Point 1: The abstract is not straightforward on the focus of the review. The main topic of the review seems to be the relevance of redox-sensitive modifications of proteins involved in MAPK and NOS signaling, but this is not evident in the abstract, since the leading sentences introduce ROS, then MAPKs and RSK1 regulation of NOS, and only then (line 16) S-glutathionylation is finally mentioned. The focus of the review would be much clearer if the introductionary sentences would describe the cysteine-based redox switches in primis, and then their role in regulation of MAPKs and NOS signaling.

Response 1: As reviewer’s comment, we have rephrased “S-glutathionylation “ to “cysteine-based redox modification” in the abstract section (see lanes 16 and 20, highlighted in red).

Point 2: Chapter 2 would be more useful if it could explain the biological meaning of protein S-glutathionylation, its impact in biology (how many proteins have been found to undergo this kind of modification?) and then the details of how it happens and how it is investigated. The focus of the review is the modification of NOS and CAMKs, but a general introduction to this would help contextualize the topic in the background knowledge on this kind of modification.

Response 2: As reviewer’s comment, we have explained the biological meaning of protein S-glutathionylation in Chapter 2-2.

2.2. The biological meaning of protein S-glutathionylation

S-glutathionylation plays an important role in protecting cysteine residues from irreversible oxidation during oxidative stress. Meanwhile, it can change the structure, function, and activities of modified proteins which affects signal transduction. Both the forward- (S-glutathionylation) and reverse- (deglutathionylation) reactions are involved in the pivotal redox-dependent cell signaling cascades. Grx is an essential thioltransferase whose primary role is to reduces S-glutathionylation. Grx 1 insufficiency results in increased levels of S-glutathionylation, implicated in many pathological events, including cardiovascular and Parkinson’s diseases. Since Grx catalyzes GSH-dependent reduction of S-glutathionylation, lower GSH:GSSG ratio might attenuate Grx activity, mimicking Grx deficiency. In contrast, Grx 1 plays a primary proinflammatory role in microglia or an anti-angiogenic role in the vasculature. More than one hundred S-glutathionylation proteins have been experimentally verified. Thus, S-glutathionylation is thought to participate in the protection and progression of pathological consequences, requiring examination of cell-type specific manipulation of Grx1 content or activity (see Chapter 2-2, lanes 151-164, highlighted in red).

Point 3: Figure 2 should indicate the difference among sulfenylation, sulfinylation and sulfonylation (label the modifications).

Response 3: As reviewer’s comment, we have indicated the difference among sulfenylation, sulfinylation and sulfonylation as; Protein sulfenylation (P-SOH) is further oxidized to generate protein sulfinylation (P-SO2H) and protein sulfonylation (P-SO3H). The former is reversible and latter is irreversible terminal modifications (see figure 2 legend, lanes 144-146, highlighted in red).

Point 4: Figures should be cited in the text but not in the subtitles. Why are the Figures and Tables in a separate section?

Response 4: As reviewer’s comment, we have cited the Figures and Tables in correct position.

Minor points:

Line 19. What ‘those’ stands for?

Response: ‘those’ stands for CaMKI and CaMKII and we have changed ’those’ to ‘that’ (see lane 19, highlighted in red).

Lines 32-33. This affirmation is quite general and sincerely redundant. Either the reader do not know MAK cascades, and the authors should explain them far better, or the authors assume the reader already know what is the matter, and this affirmation is not necessary at all.

Response: We have eliminated the sentence, as reviewer’s comment.

Line 45. A space needs to be removed

Response: We have removed a space (see lane 45, highlighted in red).

Line 55. Why ‘despite’?

Response: We have changed ‘despite’ to ‘and’ (see lane 55, highlighted in red).

Line 56. A new paragraph should be initiated

Response: We have initiated as a new paragraph as reviewer’s comment (see lane 57, highlighted in red).

Line 68. Grammar needs adjustment

Response: As reviewer’s comment, we have adjusted from “via disturb the balance of redox in vivo” to “by disturbing the redox balance in vivo” (see lane 69, highlighted in red).

Line 98. That not those

Response: As reviewer’s comment, we have changed to “that” (see lane 104, highlighted in red).

Line 170. Shouldn’t Table I be cited here?

Response: We appreciated the comment, but we described the methods to identify the cysteine-based redox molecules here in 2.3, the former 2.3. And we have cited the table 1 in the Chapter 3 and conclusion section.

Line 188. O2- is mistyped

Response: As reviewer’s comment, we have changed ‘O2–’ to ‘O2’ (see lane 192, highlighted in red).

Line 227. ‘RSKs is’ should be either ‘RSK1 is’ or ‘RSKs are’.

Response: As reviewer’s comment, we have changed ‘RSKs’ to ‘RSK1 is’ (see lane 230, highlighted in red).

Line 234, it ‘was inhibited’.

Response: As reviewer’s comment, we have rephrased as ‘it was inhibited’ (see lane 237, highlighted in red).

Line 246. Title needs fixing

Response: As reviewer’s comment, we have fixed the title (see lane 262, highlighted in red).

Line 273. Cys6 is mistyped.

Response: As reviewer’s comment, we have changed to Cys6 (see lanes 298, highlighted in red).

Line 292. The title is not explicative enough of the focus of this paragraph.

Response: As reviewer’s comment, we have changed to ‘Roles of protein S-glutathionylation in deseases.’ (see lane 317, highlighted in red).

Line 293. Who defines when glucose usage is ‘excessive’?

Response: As reviewer’s comment, we have rephrased as ‘due to its requirement for 20% of the total basal oxygen budget to support ATP intensive neuronal activity,’ (see lanes 318-319, highlighted in red).

Line 343. ‘polysulfidation’ is mistyped

Response: As reviewer’s comment, we have changed to polysulfidation’(see lane 367, highlighted in red).